# Observation of High Magnetic Bistability in Lanthanide (Ln = Gd, Tb and Dy)-Grafted Carbon Nanotube Hybrid Molecular System

**DOI:** 10.3390/ijms241512303

**Published:** 2023-08-01

**Authors:** Venkateswara Rao Sodisetti, Andreas Lemmerer, Daniel Wamwangi, Somnath Bhattacharyya

**Affiliations:** 1Nano-Scale Transport Physics Laboratory, School of Physics, University of the Witwatersrand (Wits), Johannesburg 2050, South Africa; svr.quantum@gmail.com; 2Molecular Sciences Institute, School of Chemistry, University of the Witwatersrand (Wits), Johannesburg 2050, South Africa; andreas.lemmerer@wits.ac.za; 3DSI-NRF Centre of Excellence in Strong Materials and School of Physics, University of the Witwatersrand (Wits), Johannesburg 2050, South Africa; daniel.wamwangi@wits.ac.za

**Keywords:** carbon nanotubes, lanthanide composite, magnetization, molecular magnetism, magnetic bistability, superparamagnetic behavior

## Abstract

There is an immense research interest in molecular hybrid materials posing novel magnetic properties for usage in spintronic devices and quantum technological applications. Although grafting magnetic molecules onto carbon nanotubes (CNTs) is nontrivial, there is a need to explore their single molecule magnetic (SMM) properties post-grafting to a greater degree. Here, we report a one-step chemical approach for lanthanide-EDTA (Ln = Gd^III^, **1**; Tb^III^, **2** and Dy^III^, **3**) chelate synthesis and their effective grafting onto MWCNT surfaces with high magnetic bistability retention. The magnetic anisotropy of an Ln-CNT hybrid molecular system by replacing the central ions in the hybrid complex was studied and it was found that system **1** exhibited a magnetization reversal from positive to negative values at 70 K with quasi-anti-ferromagnetic ordering, **2** showed diamagnetism to quasi-ferromagnetism and **3** displayed anti-ferromagnetic ordering as the temperature was lowered at an applied field of 200 Oe. A further analysis of magnetization (M) vs. field (H) revealed **1** displaying superparamagnetic behavior, and **2** and **3** displaying smooth hysteresis loops with zero-field slow magnetic relaxation. The present work highlights the importance of the selection of lanthanide ions in designing SMM-CNT hybrid molecular systems with multi-functionalities for building spin valves, molecular transistors, switches, etc.

## 1. Introduction

Single molecule magnets (SMM) are compounds that display magnetic properties of purely molecular origin, which are different from their bulk compounds where magnetic properties arise from collective long-range magnetic moment ordering [1]. Primarily due to their slow relaxation of magnetization and magnetic hysteresis, SMMs display interesting quantum features such as temperature-independent magnetic relaxation, butterfly-shaped hysteresis loops [2], quantum tunneling of magnetization (QTM) [3,4,5] and quantum phase interference [6], which are important for developing molecular spintronics, high-density information storage units and devices for quantum computation [7,8,9,10]. However, these quantum features are only seen at sub-kelvin temperatures. Many efforts have been expended in modulating magnetization dynamics through the coordination of molecular interactions to enable these features to be seen at higher temperatures (near liquid nitrogen temperatures—77 K) [11,12]. SMM behavior was first observed in Mn(III) compounds and attributed to a combination of Jahn–Teller distortions, ferromagnetic alignment and anti-ferromagnetic exchange interaction between Mn(III) and Mn(IV) [13,14]. This followed wide research in 3*d*, 4*f* and 3*d*–4*f* combination electronic systems looking for SMM behavior with large thermal energy barriers for magnetization reversal. Recently, the focus has shifted towards single lanthanide ion-based molecular systems, in which the SMM character is evident at a single-ion level [1]. Primarily due to their intrinsic spin–orbit coupling (*S* + *L*) and crystal field effects (CFE), lanthanide ions show large magnetic anisotropy and therefore large anisotropic energy barriers, paving the way for slow magnetic relaxation [15]. Lanthanide ion-containing molecules have thus continued to attract intensive research interest in the design and assembly of SMM molecules, mostly due to their tunable magnetic anisotropy [16,17]. Moreover, they possess interesting quantum features that could be exploited for quantum technological applications at high temperatures [18,19,20].

Contemporarily, the idea of synthesizing a hybrid molecular complex that involves grafting SMM onto carbon nanotubes and accessing the quantized magnetic state via CNT electrical conduction properties is being explored to a great extent [21,22]. Carbon nanotubes possess excellent electronic properties with low spin-orbit coupling, low hyperfine coupling and quantized phonon modes. Additionally, their diamagnetic properties make them suitable for accessing SMMs’ quantum features, which are observed in the low-dimensional realm without perturbing the magnetic properties of SMMs. Porphyrin dimeric rings with copper as the center core magnetic ions grafted onto carbon nanotubes have well-preserved spin geometry and therefore have shown quantum coherence times of 25 μs [23]. Specifically, Ln molecules based on Gd, Tb and Dy grafted onto CNTs are widely explored in molecular spintronics including spin valves [24,25], FET transistors [26] and quantum information processing [7,27,28]. These hybrid complexes, often referred to as molecular double quantum dots, have many advantages in building novel electronic devices, such as tailoring desired magnetic properties through grafting specific magnetic molecules onto CNTs [29]. Further grafting allows the introduction of spin degrees of freedom into the molecular hybrid, where new energy scales such as spin–spin exchange energy (J), spin-dependent correlations (Kondo peak at zero bias voltage) and spin–phonon coupling arise and these can be utilized for building novel spin-based molecular electronics. Recently, the spin degree of freedom introduced through the sp^3^ defect on a single-walled CNT via nitrobenzene diazonium tetrafluoroborate has shown a long coherence time of 8 μs, thus having the plausibility of quantum control operations through a one-dimensional (1D) molecular platform [30]. A detailed review of the progress that has been made on rare earth element-based SMMs can be found elsewhere [31]. However, various aspects of this supramolecular complex system, such as their magnetic bistability upon magnetic molecule attachment to CNTs, magnetic property tunability through magnetic anisotropy and large ground-state splitting through coordination chemistry, are yet to be widely explored.

Terbium stabilized with phthalocyanine anchored onto SWCNTs has shown improved magnetic properties [22]. Furthermore, spin valve devices made out of this hybrid material have shown giant magnetoresistance at low temperatures [26]. In 2018, we reported the Kondo effect with enhanced magnetic properties in a Gd-MWCNT system [25], and thereafter demonstrated how the magnetic properties could be tuned by the careful chemical functionalization of MWCNTs [32]. Later, we showed how spin relax through interaction (spin–phonon coupling) with the discrete phonon spectrum of CNTs [33]. Herein, we report the magnetic anisotropy and magnetization dynamics observed by replacing the central ion in the Ln-EDTA (Ln = Gd^III^, **1**; Tb^III^, **2** and Dy^III^, **3**)-grafted MWCNT molecular system. We have chosen Gd—4*f*^7^ (^8^S_7/2_), Tb—4*f*^8^ (^7^F_6_) and Dy—4*f*^9^ (^6^H_15/2_) as our ideal SMM systems since they display isotropic and oblate-shaped electronic distribution systems, respectively (Appendix A), which provide large magnetic anisotropies in the strong axial ligand fields and effectively gain slow magnetic relaxation. Furthermore, a 1D molecular network of these lanthanoid EDTA chelate systems, exhibiting slow magnetic relaxation, has already been reported [34,35]. Gd^3+^- and Tb^3+^-based 1D polymers have shown slow magnetic relaxation through superexchange interactions [36], while Dy^3+^ ion-based SMMs have shown a magnetization reversal barrier exceeding 2000 K [37]. However, when these molecules are grafted onto conducting surfaces, they display reduced/nullified magnetism due to extensive screening from the outer environment. On the contrary, effective grafting onto an MWCNT and the arrangement of metal SMMs into a 1D network can improve the magnetic behavior (slow relaxation) significantly through dipolar and superexchange interactions (e.g., RKKY) between the spin carriers mediated via the π electrons of CNTs. Henceforth we have chosen Gd-, Tb- and Dy-EDTA as magnetic core ions to graft onto CNTs and, more importantly, the post-grafting retainment of their magnetic bistability is studied here. Structural characterization including HR-TEM and single-crystal X-ray diffraction was conducted on all three Ln-EDTA complexes to study the magneto-structural correlations and the origin of the magnetic interactions, respectively.

## 2. Results

Ethylenediaminetetraacetic acid (EDTA) is a chelating agent that is widely used to trap metal ions for diverse industrial applications [38]. It has a strong affinity towards exophilic nature metals such as lanthanide ions; hence it can be used as an efficient metal-trapping agent. Gd-EDTA chelate magnetic properties have already been studied extensively when used as a contrasting agent in Magnetic Resonance Imaging (MRI) and further it has been reported that this chelate exhibits an unusually slow relaxation of magnetization at low temperatures [34]. Moreover, control of the assembly of the discrete 1D Ln complex on CNTs has added benefits for the advance of molecular magnetic materials. Motivated by this, we synthesized Ln-EDTA (Ln = Gd, Tb, Dy) chelates and grafted them onto MWCNTs through a one-step chemical method. Single crystal X-ray diffraction measurements were carried out to understand the molecular structure and correlate the magnetic properties of the final Ln^3+^-MWCNTs found from the DC-VSM measurements.

### 2.1. Crystal Structure Description

Ln-EDTA chelate is prepared via the solvothermal synthesis method, followed by slow evaporation of the solvent under ambient conditions to obtain high-quality single crystals. The crystal structure of Ln-EDTA complexes shown in Figure 1a,b forms a one-dimensional chain structure in the chiral space group Fdd2 with consistent cell parameters a, b and c. Detailed crystallographic information is presented in Table 1. The Ln atom is coordinated by two EDTA molecules through the Ln-O (atoms O1, O3, O5 and O7) and the Ln-N bonds (atoms N1 and N2), as shown in Figure 1a, through the carboxylate and amine groups of the EDTA, respectively. It is further coordinated by two O atoms, O10 and O11, which bond with Na atoms Na3 and Na4, as well as by O3 bonding to Na4. Ln has one coordinated water molecule and Na has also a water molecule bonded to it. Furthermore, Na4 bonds to O9 and Na1 and Na3 bonds to Na2. This is the connectivity in the asymmetric unit. Overall, the bonding results in a 1D chain network within a 3D packing arrangement (Appendix A).

Intramolecular distances and intermolecular distances between Ln atoms play a significant role in magnetic relaxation dynamics. Here, we find that the intramolecular distances (Tb…Tb distance is 6.071 Å) likely influence the interactions between the spin carriers, in contrast to the interactions between the chains (Tb…Tb distance is 9.437 Å) (see Appendix A). The packing diagram (Appendix A) shows that the Tb- and Dy-EDTA form an overall 3D packaging arrangement; however, distortion within the individual Ln metal chain network can vary differently and can significantly affect the overall magnetic anisotropy.

### 2.2. Grafting Ln-EDTA SMMs onto Carbon Nanotubes

Before the grafting of the Ln-based SMMs, the MWCNTs were pretreated in an acid mixture of concentrated sulfuric acid and nitric acid (H_2_SO_4_/HNO_3_—3:1 ratio); this ensures the removal of catalytic metal particles (Fe) and the introduction of carboxylic groups to the surface of the nanotubes. Traditionally, two synthetic chemical routes were employed in attaching SMMs to the CNT surface; first, covalent bonding between the functionalized CNTs and the molecular complex; and second, non-covalent attachment via electrostatic interactions and/or *π*–*π* stacking of molecular complex on the CNTs. It is envisaged that the non-covalent attachment of the molecules preserves the integrity and electronic character of the carbon nanotubes. There are, however, studies reporting the successful covalent attachment of molecules to CNTs without hindering the electronic properties of the CNTs [39] through a careful degree of acid treatment during CNT functionalization [32]. Furthermore, it is interesting to study the surface interactions between CNTs and metal ions stabilized with EDTA ligand. Here, we have employed a covalent attachment of Ln-EDTA molecules that involves treating –COOH-terminated MWCNTs with Ln-EDTA molecules dissolved in DI water under continuous magnetic stirring.

Figure 2a illustrates the attachment of the Ln^3+^-EDTA molecules onto the CNTs, while Figure 2b,c and d show the HR-TEM images of Gd^3+^, Tb^3+^ and Dy^3+^ ions attached onto the nanotube surface, respectively. In general, both the covalent and non-covalent approaches can yield a large number of nanoparticles being grafted per unit volume of CNT. However, for a spintronic application, a controlled number of SMM attachments, yielding a diluted magnetic system, is essential. In our work, we have employed a careful acidic pre-treatment of the MWCNTs for a shorter duration, yielding carboxylic group terminated MWCNTs without damaging the structural integrity (see Figure 2b) and further allowing us to graft a controlled number of Ln-EDTA molecules onto the MWCNTs (Figure 2e–g show respective EDAX profiles showing Ln^3+^ ion concentration). The EDAX results show that 0.5 wt%, 0.8 wt% and 1 wt% of Gd, Tb and Dy, respectively, were doped onto the MWCNTs. This further allows us to study the single-molecule magnetic behavior of the respective lanthanide molecules.

### 2.3. Magnetization Studies of Ln-EDTA-Grafted CNT Molecular System

Temperature-dependent direct current (dc) magnetic susceptibility measurements from 2–300 K with an applied dc field of 200 Oe were carried out to study the static magnetic behavior of the Ln-EDTA grafted MWCNTs. Zero-field cooled (ZFC) magnetic measurements reveal a one-dimensional hybrid molecular system displaying high magnetic anisotropies (Figure 3a and Figure 4a,b). The molar magnetic susceptibility ($\chi_{m}$) of the **1**-molecular system (Figure 4a) showed a gradual decrease in the temperature range from 300 to 70 ± 4 K and thereafter took negative values of $\chi_{m}$ to 2 K as the temperature was lowered (Figure 4b), signaling a quasi-anti-ferromagnetic interaction between the unpaired spins of the Gd^3+^ and the conduction electrons of the CNTs. On the contrary, the **3**-molecular system (Figure 4b) showed a linear increase in $\chi_{m}$ with temperature in the range from 300 to 20 ± 3 K and a sharp upturn to 2 K, while the **2**-molecular system (Figure 4a), $\chi_{m}$ remained constant from room temperature to 40 ± 3 K, with a modest downturn to 2 K upon a decrease in the temperature, respectively. The low-temperature decreases in the $\chi_{m}$ for both the 1- and 2-molecular systems may be due to the following possibilities: anti-ferromagnetic ordering, Kondo screening of iterant conduction electrons, thermal depopulation of the Stark sublevels, etc. As mentioned, the replacement of the central metal ions in the Ln-MWCNT hybrid molecular system led to changes in the bonding configurations and subsequently to varied magnetic properties. The origin of such large magnetic anisotropies can be traced back to the exchange interactions between the metal ion and the π-electronic systems of the MWCNTs. The local spins (molecular magnets) with a proximity to the itinerant conduction electrons of the nanotubes develop either a ferromagnetic or an anti-ferromagnetic ordering through a variety of exchange interactions including Coulomb, spin–exchange [33] and weak intra- or inter-molecular interactions [40]. Further, competition between these interactions is mediated by the Ruderman–Kittel–Kasuya–Yosida (RKKY) interaction. The effective RKKY exchange interaction ($J_{RKKY}$) dictates the magnetic ordering in a one-dimensional molecular system.

To further understand the magnetic behavior of these molecular systems, inverse susceptibility as a function of temperature ($\chi_{m}^{-1}$ vs. *T*) was analyzed (Figure 3b and Figure 4a,b). An interesting magnetic reversal future is seen for the **1**-molecular system, indicating a magnetic phase transition from diamagnetic to quasi-anti-ferromagnetic appearing around transition temperature *T* ≅ 70 ± 4 K. It can be noted that $\chi_{m}^{-1}$ has a crossover from positive to negative magnetic moments as the temperature is lowered. A similar magnetization reversal observation was previously made in rare-earth ion-modified carbon nanotubes [41]. The plausible explanation for this phenomenon is competition and balance between thermal activation relaxation and macroscopic quantum tunneling. As the temperature is lowered, the thermally activated relaxation becomes slow and the macroscopic quantum tunneling-based relaxation becomes predominant [42]. Moreover, the CNT shape anisotropy (length/diameter ratio) also influences the interactions between the grafted magnetic ions. Given the tubular geometry of CNTs, the shape can influence magnetization reversal through coherent rotations and the transverse rotation of unpaired electron spins, and as well through the nucleation and propagation of the vortex domain walls, which influence the spin rotations [43,44].

The inverse magnetic susceptibility ($\chi_{m}^{-1}$ vs. *T*) of the **2**-molecular system remains constant up to 35 ± 2 K and thereafter a subtle increase in χ**^−^**^1^ with decreasing temperature is observed. This is characteristic of a typical quasi-ferromagnetic behavior. Thus, the Tb-MWCNT system undergoes a symmetry-breaking transition from a disordered to a magnetically ordered state. At higher temperatures, thermal fluctuations are relatively stronger than the magnetic ion (local spin) interactions. However, when lowering the temperature, at the onset of a certain temperature (here *T* ≅ 35 ± 2 K), the interaction between the neighboring ions starts growing stronger and magnetic ordering is developed. On the contrary, the **3**-molecular system $\chi_{m}^{-1}$ vs. *T* shows a linear decreasing trend and undergoes a sharp decrement in its $\chi_{m}^{-1}$ at 20 ± K upon lowering the temperature, which is characteristic of anti-ferromagnetic ordering. Here, an extensive screening of electrons from a diamagnetic carbon nanotube yields a net reduced magnetic moment and hence develops an effective anti-ferromagnetic ordering. Curie–Weiss law fitting is conducted for the **3**-molecular system to calculate the effective magnetic moment.
(1)$$\chi_{M}=\frac{C}{T-\theta }$$

Here, θ is the Weiss constant and *C* is the Curie constant, found to be −150 ± 4 K. It is noteworthy that in both the **1**- and **2**-molecular systems $\chi_{m}^{-1}$ vs. *T* was not able to fit with the Curie–Weiss law since local moments are influenced by factors such as low one-dimensionality, core diamagnetism from the CNTs, etc.

Figure 5a,c,e show the isotherm plots depicting the variation of magnetization with applied magnetic field (M vs. H) plots for the Gd^3+^-, Tb^3+^- and Dy^3+^-MWCNT molecular systems, respectively. Magnetization hysteresis measurements were collected in the temperature range of 2–300 K. Interestingly, all three molecular systems showed characteristic “S”-shaped hysteresis with an open loop at zero field indicating a slow magnetic relaxation. This hysteresis loop and the area enclosed within it grew larger with a temperature decrease to 2 K, which indicates the characteristics of a quantum-tunneling mechanism at low temperatures, which is typical behavior of SMMs [45]. The coercivity ($H_{c}$), remnant magnetization ($M_{r}$) and saturation magnetization ($M_{s}$) of the respective molecular system at temperatures 2 K and 300 K can be found in Table 2. It is worth noting that the presence of significant remnant magnetization and coercive field at 2 K for all three molecular systems attests to the magnetic bistability retainment post-grafting onto MWCNTs. Magnetic bistability is an important characteristic of SMMs and is dependent on ground state spin (*S*) and zero-field splitting parameter (D) with a negative value. Often the superexchange interactions between the magnetic ions combined with magnetic anisotropy results in a large energy barrier for magnetization reversal between the two magnetically polarized states (negative and positive moment). Slow relaxation in the magnetization is the result of this large energy barrier formation and this property can be used for high-density data storage applications. It is worth noting that, in our work, all three compounds showed a hysteresis loop at low temperatures; more considerably the **2** and **3** molecular hybrids showed a large hysteresis loop, respectively (see Table 2). The reason for gaining large hysteresis in the magnetization is discussed in Section 4.

To further evaluate the hysteresis loops seen in the respective Ln^3+^ ion-doped MWCNTs, the derivative magnetization vs. field (d*M*/d*H*) of all three molecular hybrids was analyzed (Figure 5b–f). First, the field is ramped to 20 k Oe, where all the doped molecules’ magnetic moments are aligned parallel to the external field and magnetization saturation is achieved. When the field is swept back to zero from 20 k Oe, the magnetization should be zero in a case where there are equal numbers of “spin up” and “spin down” molecules. However, in all three MWCNT hybrid molecular systems, step-like futures in the hysteresis are seen at 600 Oe for **1**, 1040 Oe for **2** and 1090 Oe for **3,** respectively, on both sides (positive and negative) of the applied field at 2 K. This indicates that there is an energy barrier for the molecules to reverse magnetization when the external field is swept in cycles between positive and negative fields. Further, it can be observed that these step-like positions concerning the magnetic field vary for **1**, **2** and **3**, which correlates with the varied relaxation energy barrier due to varied magnetic anisotropy by replacing the central metal ion in the Ln-EDTA and effective exchange interactions with the π electrons of the MWCNTs. It also becomes evident through the d*M*/d*H* plots that slow relaxation is absent at higher temperatures. It can be noted that the step-like sharp peaks are absent and merge as a single sharp peak seen at zero field, indicating a faster relaxation of magnetization at temperatures > 20 K (Inset Figure 5b–f).

Field-dependent magnetization (M vs. H), shown in Figure 5a,b for molecular systems **1** and **2**, shows magnetization curve saturates at 19 k Oe and 10 k Oe, respectively, and remains constant at low temperature and high field. On the contrary, for molecular system **3** (Figure 5c), the magnetization curve at 2 K increased till 6 k Oe, remained constant to 15 k Oe and showed a decreasing trend at higher fields thereafter; a reasonable explanation is that there is a large diamagnetic influence by the MWCNTs. Diamagnetic materials such as graphite/graphene have no unpaired electrons and hence no net magnetic moment per atom, and therefore have a large diamagnetic susceptibility [46]. Electrons present within the graphitic layers of the MWCNTs can rearrange their orbits, and therefore can create small eddy currents that oppose the external magnetic field [47]. Figure 5c shows a slight decrease in magnetization at 2 K at higher fields and this can be attributed to the destruction of eddy currents. However, the diamagnetic contribution is greatly reduced at higher temperatures, and this can be ascribed due to Van Vleck’s paramagnetic contributions from localized defects in the MWCNTs [48,49]. As mentioned above, a single Ln-MWCNT molecular system with a replaced central metal ion shows significant magnetic anisotropic behavior.

The plots of M vs. HT^−1^ for the Gd-, Tb- and Dy-MWCNT systems are shown in Figure 6a,c,e*,* respectively. It is worth noting that the M vs. HT^−1^ curves for all three molecular systems do not superimpose onto a single master curve, confirming the presence of magnetic anisotropy. To obtain further complementary information on the Ln-MWCNT hybrid molecular complex, we performed M/M_s_ plot fitting with theoretical Brillouin curves for different values, respectively (Figure 6b,d,f). Magnetization concerning the field can be fitted using the following equation:(2)$$M=ngJ\mu_{B}B_{J}\left(x\right)$$
where ***n*** is the number of Ln^3+^ ions in the sample, ***g*** is the electron spin factor, ***J*** is the total angular momentum, $\mu_{B}$ is the Bohr magneton and the Brillouin function is
(3)$$B_{J}\left(x\right)=\frac{2J+1}{2J}\coth\left(\frac{2J+1}{2J}x\right)\;-\frac{1}{2J}coth\left(\frac{1}{2J}x\right)$$
and
(4)$$x=\frac{gJ\mu_{B}H}{k_{B}T}$$
where $k_{B}$ is the Boltzmann constant.

Finally, an effective magnetic moment can be derived by substituting the ***J*** value in Equation (5) as follows:(5)$$\mu_{eff}={g\left[J\left(J+1\right)\right]}^{1/2}\mu_{B}$$

We considered the best-fitting values and found *J* = 8 and *J* = 7.4 for molecular systems **2** and **3,** respectively. However, we could not fit the molecular system **1** and this might be due to the presence of metal impurities or to defective sites opened on the CNT surface during acid treatment. Following Equation (5), we find the effective magnetic moment $\mu_{eff}$ to be 12.7279 $\mu_{B}$ and 10.4859 $\mu_{B}$ for systems **2** and **3**, respectively. These experimentally derived values were compared with the theoretical values $\mu_{eff}$ = 9.72 $\mu_{B}$ and 10.65 $\mu_{B}$; we found that the Tb-MWCNT showed a higher value and the Dy-MWCNT system showed a lower value (Table 3). This further confirms the evaluation of inverse susceptibility where 2 displays ferromagnetic and **3** shows anti-ferromagnetic ordering.

## 3. Discussion

The introduction of magnetic molecules onto a conducting surface such as a CNT will enable accessing spin degrees of freedom where condensed matter phenomena interactions such as spin–spin exchange, spin–phonon coupling and spin correlations can be realized to build a variety of molecular electronic devices. Through careful control of the chemical functionalization of CNTs with magnetic molecules, we can realize enhanced magnetic properties in the compound molecule. All three magnetic ions, **1**, **2**, and **3**, were grafted onto the CNTs via EDTA ligand, respectively, and the controlled attachment led to the grafting of 0.5–1 wt% of magnetic metal ions onto the CNT surface, confirmed through EDAX results.

Since the concentrations of Ln^3+^ ions on the CNT surface are lower, it can be inferred that the CNT hybrid molecular system is grafted with isolated Ln^3+^ ions rather than a bulk chain of molecular magnets. Therefore, the enhanced magnetic properties observed are due to superexchange interactions. Furthermore, the steps seen in Figure 5b,d,f are indicative of these interactions observed only at low temperatures (<10 K). A similar kind of magnetization phenomena is observed in diluted Dy(III) SMMs [50] and Tb(III)-encapsulated SWCNTs [51]. It is interpreted that grafting affects the magnetic dipole interactions between the neighboring Ln^3+^ ions, which effectively reduces the QTM-type relaxation and therefore gains slow relaxation.

Table 3 shows the effective energy barrier of the respective magnetic molecule coupled CNT hybrid molecular systems. It should be noted that a significant magnetic anisotropy is shown in the Ln-EDTA-coupled CNT molecular hybrid by replacing the central metal ion. The plausible explanation for the enhanced and varied magnetic properties is due to complex magnetic interactions between the iterant conduction electrons of the CNTs and rare earth ions. It is noteworthy to mention that both systems **2** and **3** gained a magnetization reversal energy barrier ($\mu_{eff})$ of 12.7279 $\mu_{B}$ and 10.4859 $\mu_{B}$, respectively, due to the slow magnetic relaxation via the Orbach mechanism (suppressed QTM). In contrast, for system **1,** we could not fit the experimental data with the Brillouin function (using Equation (3)) since QTM-type relaxation is dominant. A theoretical model was developed in understanding the different dominant relaxation mechanisms involved when an SMM is attached to a conducting surface [52]. Particularly when a magnetic spin center is grafted onto a conducting surface, the magnetization manifestation can be due to spin-flipping events assisted by the phonon spectrum of the CNT. Further understanding of the magnetic mechanism in this molecular hybrid will be studied through frequency-dependent AC magnetization and low-temperature transport studies.

## 4. Materials, Methods and Instrumentation

All reagents were purchased from Sigma Aldrich and were used without further purification. Carbon nanotubes were grown using the catalytic chemical vapor deposition technique at the Nano-Scale Transport Physics Laboratory (NSTPL), School of Physics, University of the Witwatersrand. MWCNTs were further subjected to acidic treatment to remove the metal impurities and to promote hydrophilic nature.

### 4.1. Ln–EDTA Chelate Synthesis

Ln-EDTA chelates were synthesized through the solvothermal evaporation method. First, 0.5 mM (0.146 g) of ethylenediaminetetraacetic acid (EDTA) was dissolved in 25 mL of DI water placed in a 250 mL round-bottomed flask. To this solution, 0.5 mM of Ln (Ln = GdCl_3_.6H_2_O (0.186 g), **1**, DyCl_3_.6H_2_O (0.188 g), **2** and TbCl_3_.6H_2_O (0.186 g), **3**, respectively) was added. 1 M of NaOH was added dropwise to promote ligand solubility. The suspension was stirred and heated to reflux for 3 h in an oil bath at a temperature of 70 °C assisted with a magnetic stirrer with a hot plate. After the addition of NaOH and within the first hour of stirring, the white-colored suspension started turning transparent. The suspension was filtered with a PVDF membrane (0.22 μm) using a reduced pressure technique. The collected suspension was left out to precipitate for the next 30 h. Ethanol was spread as a layer to promote the formation of crystals. The crystals collected were further resuspended in a 25 mL DCM solution to further remove the impurities.

### 4.2. Ln–EDTA Grafting onto MWCNTs

Functionalized MWCNTs with pre-determined carboxylic (-COOH) group percentage through Boehm titration and Ln-EDTA chelate were weighed stoichiometrically in the ratios of 1:1 and were mixed in 50 mL of DI water. The mixture solution was magnetically stirred at room temperature for a period of 24 h with the intermittent step of ultrasonication to ensure the proper dispersion of MWCNTs across the liquid medium. The resultant mixture was subjected to centrifugation and decant was removed before filtering with a 0.22 μm PVDF membrane filter. Finally, the solid was vacuum dried under reduced pressure at a temperature of 100 °C for 6 h.

### 4.3. HR–TEM and EDAX

High-resolution transmission electron microscopy (TEM) was performed using JEOL JEM-2100 Plus operating at 200 kV accelerating voltage. Samples were prepared by casting drops of Ln-EDTA-grafted CNT hybrid molecular material suspended in Ethanol solvent onto a copper grid and left to dry in air.

### 4.4. Single-Crystal X-ray Diffraction

Intensity data were determined on a Bruker Venture D8 Photon CMOS diffractometer with graphite-monochromated MoKα_1_ (λ = 0.71073 Å) radiation at 173 K using an Oxford Cryostream 600 cooler. Data reduction was carried out using the program SAINT+, version 6.02, and empirical absorption corrections were made using SADABS. Space group assignment was made using *XPREP*(Version 2014/2). The structure was solved in the *WinGX*(Version 2021.3) Suite of programs, using intrinsic phasing through *SHELXT*(Version 2018/2) and refined using full-matrix least-squares/difference Fourier techniques on F using *SHELXL-2017*(Version 2019/3). All C-bound hydrogen atoms were placed at idealized positions and refined as riding atoms with isotropic parameters 1.2 times or 1.5 times those of their parent atoms. All O-bound hydrogen atoms were in the difference Fourier map and refined as riding on their parent atoms with isotropic parameters 1.5 times those of their parent atoms. Diagrams and publication material were generated using *ORTEP-3*(Version 2020.1) and *PLATON*(Version 130720).

### 4.5. DC PPMS Magnetometry

Low-temperature magnetic measurements were carried out using a DC-VSM (DynaCool 12T PPMS) magnetometer between the temperatures 2 K and 300 K at an applied field of 200 Oe. ZFC measurements were carried out by first cooling the sample from room temperature to 2 K under a zero magnetic field and later magnetization data were acquired in a warming cycle after applying a magnetic field of 200 Oe. FC measurements were carried out by cooling the sample from 300 K to 2 K at a magnetic field of 200 Oe and magnetization data were acquired during the warming cycle while keeping the field on.

The resolution of the DynaCool 12 PPMS in the measurement of the magnetic moment is 0.5 × 10^−7^ emu. Both ZFC/FC and M vs. H measurements were conducted twice and the data collected were averaged to avoid discrepancies in the data, therefore increasing the confidence interval level of the data collected from the measurements.

## 5. Conclusions

SMMs based on lanthanide ions show high anisotropy due to the local geometry and symmetry. The magnetic properties of the SMMs can be tuned by varying the lanthanide centers with the same ligand environment. Here, we have described a one-step approach for Ln-EDTA chelate synthesis and the controlled attachment of 4f-electronic system-based SMMs (Ln = Gd, Tb, Dy) onto MWCNTs with a higher-order retainment of magnetic bistability. We have shown that intramolecular distances between the Ln-Ln interactions are dominant in an SMM 1D chain network and influence the spin–spin interactions, effectively influencing the magnetic properties. However, when introduced as isolated Ln^3+^ magnetic impurities onto a conduction channel such as a CNT, the superexchange interactions via the RKKY mechanism and extensive π-electron screening will influence the final magnetic properties. Molecular system **1** (Gd**^3+^**-MWCNT) showed superparamagnetic behavior, while **2** and **3** (Tb^3+^- and Dy^3+^-MWCNT systems) showed a clear characteristic of SMM behavior. It is noteworthy that system **1** changes its magnetization value from positive to negative as the temperature is lowered. Furthermore, we have shown that the characteristic properties of individual SMMs can be effectively transferred to the CNT hybrid molecular system through a careful grafting approach. Low-dimensional nanoparticles must show enhanced magnetic properties and the same is realized for all the Ln^3+^-grafted CNT hybrid molecular systems: **1** ($H_{c}$ = 250 Oe), **2** ($H_{c}$ = 320 Oe) and **3** ($H_{c}$ = 322 Oe). Our work shows the importance of central ions in the hybrid molecular complex and their effect on magnetic properties; therefore, we envisage careful engineering of a molecular system that can allow us to tune desired magnetic properties. It is important to design an SMM molecular system where the individual spin center can be accessed and regulated easily. In this regard, our work provides insights into developing a simple approach to effectively grafting spin centers onto a diamagnetic material and accessing them via the semiconducting nature of CNTs.

## Figures and Tables

**Figure 1 ijms-24-12303-f001:**
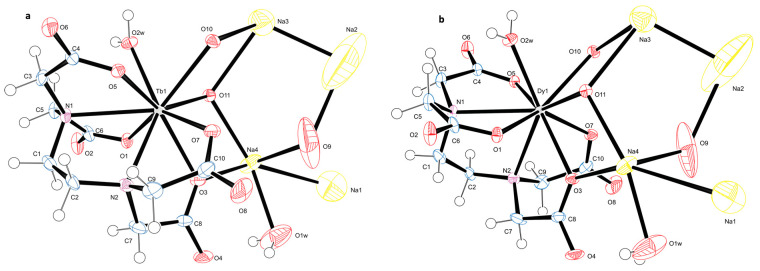
Perspective view of (**a**) Tb-EDTA and (**b**) Dy-EDTA showing the atom numbering scheme. Displacement ellipsoids are drawn at the 50% probability level.

**Figure 2 ijms-24-12303-f002:**
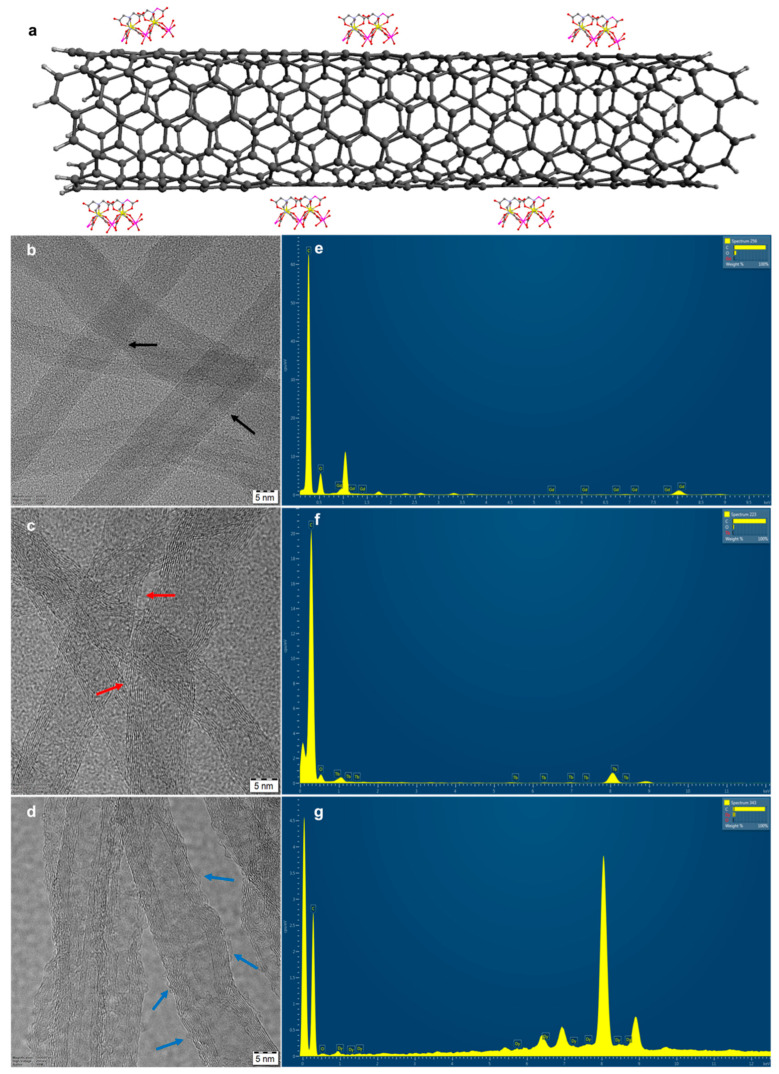
(**a**) Illustration of SMMs grafted onto carbon nanotubes. (**b**–**d**) HR-TEM images of Gd-, Tb- and Dy-EDTA magnetic molecules grafted onto MWCNTs; arrows are used to highlight the magnetic molecules grafted onto the nanotube. (**e**–**g**) EDAX showing the respective Ln ^3+^ magnetic ion concentration in the functionalized MWCNT samples.

**Figure 3 ijms-24-12303-f003:**
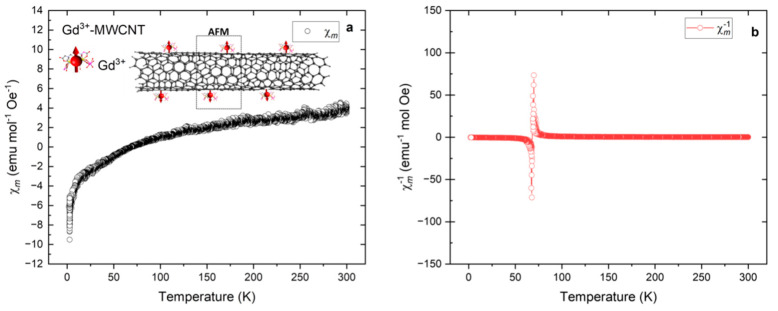
(**a**) Molar magnetization of Gd–MWCNT hybrid molecular system showing negative susceptibility at low temperatures (<70 K). (**b**) Gd-MWCNT molecular system showing significant magnetic phase change observed through crossover future from positive to negative values in inverse susceptibility ($\chi_{m}^{-1}$) at temperature *T* ≅ 70 ± 4 K.

**Figure 4 ijms-24-12303-f004:**
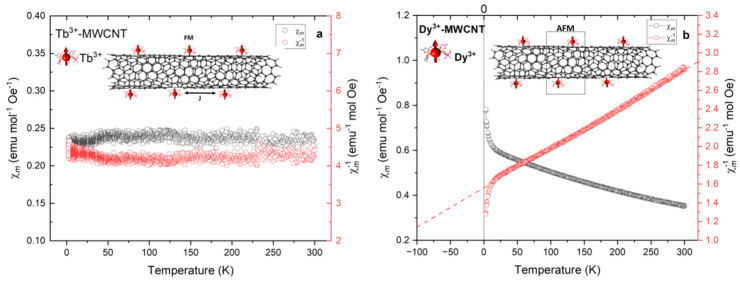
Molar magnetic susceptibilities at Zero Field Cooling of CNT molecular systems 2 and 3: (**a**) χ**^−^**^1^ vs. *T* plot of Tb^3+^-MWCNT displaying a quasi-ferromagnetism and (**b**) Dy3^+^-MWCNT supramolecular system displaying an anti-ferromagnetic ordering as the temperature is lowered.

**Figure 5 ijms-24-12303-f005:**
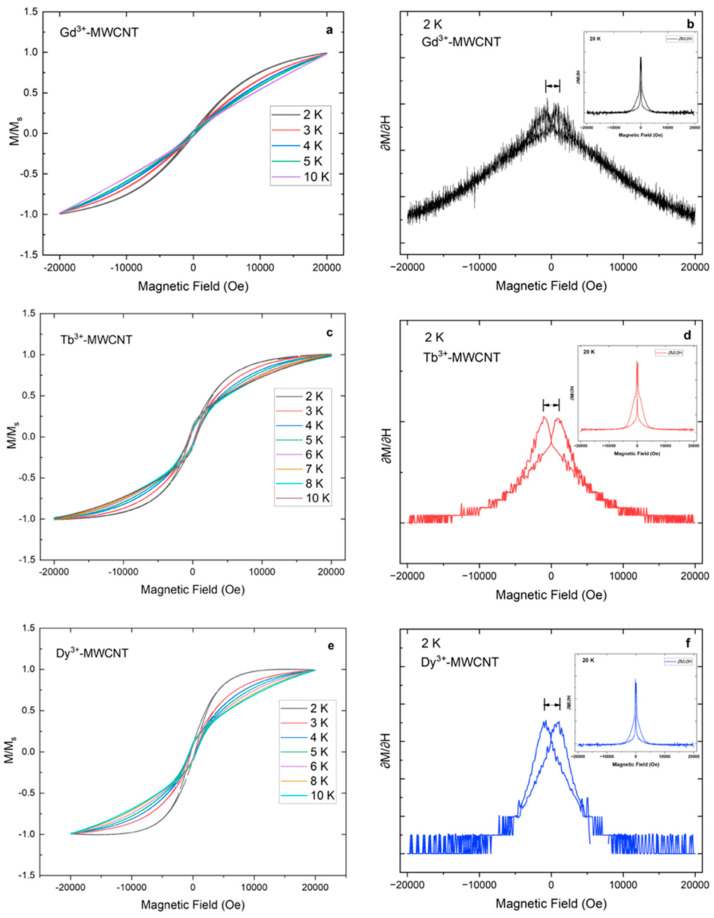
Field dependence of normalized magnetization (M/M_s_ vs. H) of Ln^3+^ (Ln = Gd (**a**), Tb (**c**), Dy (**e**))-grafted MWCNT hybrid molecular system displaying characteristic “S”-shaped hysteresis loop. Their respective behavior can be traced through d*M*/d*H* vs. H plots (Ln = Gd (**b**), Tb (**d**), Dy (**f**)).

**Figure 6 ijms-24-12303-f006:**
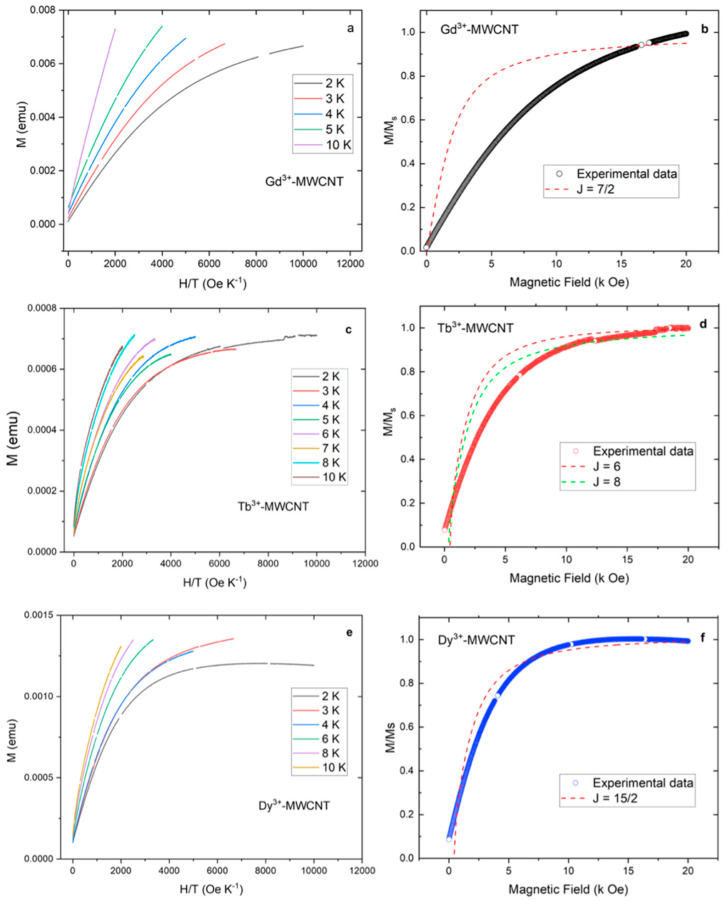
(**a**,**c**,**e**) shows magnetization vs. ratio of field and temperature (M vs. H/T) of Gd-, Tb- and Dy-grafted MWCNT systems, respectively. (**b**,**d**,**f**) Brillouin function fitting for the respective molecular systems is carried out to evaluate the exchange interaction parameter *J*. plots.

**Table 1 ijms-24-12303-t001:** Crystal data for Tb-EDTA and Dy-EDTA magnetic molecules.

	Tb-EDTA	Dy-EDTA
Formula	C_10_H_16_TbN_2_Na_4_O_13_	C_10_H_16_DyN_2_Na_4_O_13_
M_r_	623.13	626.71
Temperature/°C	−100	−100
Crystal size/mm	0.684 × 0.502 × 0.487	0.549 × 0.393 × 0.188
Crystal system	Orthorhombic	orthorhombic
Space group (no.)	*F*dd2	*F*dd2
*a/*Å	19.3067 (11)	19.3107 (18)
*b*/Å	35.1670 (19)	35.216 (3)
*c*/Å	12.0522 (6)	12.0558 (11)
*V*/Å^3^	8182.9 (8)	8198.6 (13)
*Z*	16	16
*D_c_*/g cm^−3^	2.028	2.031
*μ* (Mo-K_α_)/mm^−1^	3.608	3.796
Theta range/°	3.134 to 29.999°	2.641 to 29.996°
Total reflections	55,841	44,545
No. unique data	5936	5946
*R* (int)	0.0194	0.0194
No. data with *I* > 2σ(*I*)	5930	5933
final *R* (*I* > 2σ(*I*))	0.0230	0.0282
final *wR2* (all data)	0.0591	0.0677
Restraints/parameters	1/274	1/274

**Table 2 ijms-24-12303-t002:** Coercivity ($H_{c}$), remnant magnetization ($M_{r}$), saturation magnetization ($M_{s}$) of the Ln^3+^-MWCNT (Ln = Gd, Tb, Dy) molecular system at 2 K and 300 K.

Molecular System	$H_{c}$		$M_{r}$		$M_{s}$	
	2 K	300 K	2 K	300 K	2 K	300 K
Gd3+-MWCNT	250	145	0.026	0.010	1.307	0.022
Tb3+-MWCNT	320	58	0.073	0.012	0.44	0.054
Dy3+-MWCNT	322	65	0.084	0.011	0.27	0.025

**Table 3 ijms-24-12303-t003:** Comparison of theoretical and experimental effective energy barrier ($\mu_{eff}$).

Rare Earth in Complexes	S	L	J	g	$\mu_{eff}\;=\;g\sqrt{J(J+1)}\;\mu_{B}$	$\mu_{eff}$ Experimental
Gd^3+^ 4f^n^; *n* = 7	7/2	0	7/2	2.00	7.94	NA
Tb^3+^ 4f^n^; *n* = 8	3	3	6	1.50	9.72	12.7279 $\mu_{B}$
Dy^3+^ 4f^n^; *n* = 9	5/2	5	15/2	1.33	10.65	10.4859 $\mu_{B}$

## Data Availability

The data presented in this study are available upon reasonable request to Somnath Bhattacharyya.

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
