# Peer review of "Observation of High Magnetic Bistability in Lanthanide (Ln = Gd, Tb and Dy)-Grafted Carbon Nanotube Hybrid Molecular System"

_ijms, 2023, doi:10.3390/ijms241512303_

Round 1
Reviewer 1 Report
Dear authors,
thank ypu for your work. Please, remake your text without big loсuses (pages 5 and 11). Also I think Materials and Methods will be better to make before Resuls. In the methods, add the confidence interval and measurement error.
no comments
Author Response
Reviewer 1:
Q1. Please, remake your text without big loсuses (pages 5 and 11).
- This has been resolved now by rearranging the text.
Q2. I think Materials and Methods will be better to make before Results. In the methods, add the confidence interval and measurement error.
- The authors would like to thank the reviewer for his/her recommendation. The Materials, Methods, and Instrumentation section is moved forward and is now placed in section 2 of the manuscript.
DC magnetization measurements were conducted twice, and the data collected is averaged out to increase the confidence interval level. Additionally, the resolution of the DynaCool 12 PPMS in the measurement of the magnetic moment is 0.5´10-7 emu, the measurements in Figure 4a were taken in short applied field intervals. Similar remarks have been added to section 2.5.

Reviewer 2 Report
Ref_comments to the paper titled as “Observation of high magnetic bistability in Lanthanide (Ln = Gd, Tb, and Dy) grafted carbon nanotube hybrid molecular
system” written by the authors: Venkateswara Rao Sodisetti, Andreas Lemmerer, Daniel Wamwangi and Somnath Bhattacharyya.
It is well known that the magnetization effect can be efficiently used in the optoelectronics, atomic industry, laser techniques and in the biomedicine as well. Moreover, namely some atoms from the lanthanides group are good candidate for this aim, namely to activate the magnetization of the matrix systems. From this point of view the current article is actual and modern.
For the first, the authors have made good literature search, analyzing of 44 papers in the studied area. It is good! But, so many papers written by the last 10 years are analyzed. Please add 5-7 papers written in this area by the last 3 years.
The synthesis process and the instrumentation involved to study are useful. The mathematical procedure to explain the magnetization effect is not contradicted with our basic physical-chemical knowledge. But, among the different predicted characteristics the author have used only own data. The discussion part is so little. Please add in your table 3 - Comparison of theoretical and experimental effective energy barrier – the data observed by other scientific teams for the comparison.
Please used Latin symbols (for example, the temperature symbol) as the tilted ones.
Well, the paper is interesting and useful for the readers. Conclusion can be extended a little bit as well.
As for my local opinion, this paper can be published after the minor corrections.
Author Response
Reviewer 2:
Q1. For the first, the authors have made a good literature search, analyzing 44 papers in the studied area. It is good! But so many papers written in the last 10 years are analyzed. Please add 5-7 papers written in this area in the last 3 years.
- Appropriate supporting literature in the field of SMM and SMM functionalized CNTs from the past 3 years has been added to the manuscript. Please find the reference list detailed below.
Q2. The mathematical procedure to explain the magnetization effect is not contradicted with our basic
physical-chemical knowledge. But, among the different predicted characteristics the author have used only own data. The discussion part is so little. Please add in your table 3 - Comparison of theoretical and experimental effective energy barrier – the data observed by other scientific teams for the comparison.
- We would like to thank the reviewer sincerely for highlighting the need for data comparison. The discussion part is modified and updated. We have discussed the origin of slow relaxation in the magnetic hysteresis observed in the Ln3+ grafted CNT molecular system. Furthermore, such slow relaxation is due to the super-exchange interactions and the suppression of Quantum Tunneling Magnetization (QTM), the same has been supported by appropriate literature [50-51].
Q3. Please used Latin symbols (for example, the temperature symbol) as the tilted ones.
- Latin symbols were used wherever necessary. We thank the reviewer for highlighting this and allowing us to correct it.
Q4. Well, the paper is interesting and useful for the readers. Conclusion can be extended a little bit as well.
- The conclusion has been paraphrased and extended to make the compressive understanding of our work to the readers.

Reviewer 3 Report
The paper is devoted for investigations the magnetic bistability in lanthanide grafted carbon nanotube hybrid molecular system. The topic is generally interesting, however the paper contain unexplained places (below) and need major revisions.
The aim of the paper should be clearly formulated.
All abbreviations should be explained by first using, for example line 15 - CNT.
References to new publications (since 2020) should be added to the list of references in order to show the novelty of investigations. More comparison of obtained results with results already published in literature should be added in paper parts 2-4.
Why the data in Fig.4a is very scattered? What is the measurements accuracy of your measurements?
Investigations of additional experimental techniques like XPS or EPR should be performed for your samples.
Conclusions should be rewritten in more informative way.
Author Response
Reviewer 3:
Q1. The aim of the paper should be clearly formulated.
- Our work is to understand the retention of magnetic bistability of lanthanide ions post their grafting on the conducting surface. Furthermore, what happens when the central metal ion in the Ln-EDTA grafted CNT molecular system is replaced with ions that have varied charge distribution. Does the magnetic properties are enhanced or reduced? Such understanding will allow us the choice of metal ions to obtain specific magnetic properties. The same has been conveyed in the abstract and the Introduction. We have also updated the introduction section with recent literature on SMM functionalized CNTs, therefore conveying our aim of the work.
The authors would like to thank the reviewer for his/her recommendation.
Q2. All abbreviations should be explained by first using, for example, line 15 - CNT.
- Abbreviations are explained appropriately. Most of the abbreviations are introduced in the first section.
Q3. References to new publications (since 2020) should be added to the list of references to show the novelty of the investigations. More comparison of obtained results with results already published in literature should be added in paper parts 2-4.
- Appropriate recent supporting literature related to SMM and SMM grafted CNT systems from the past 3 years have been added to the manuscript both in the introduction and discussion sections. The same list can be found below.
Q4. Why the data in Fig.4a is very scattered? What is the measurements accuracy of your measurements?
- The change in magnetization as a function of temperature is very minimal and the data is plotted for (emu mol-1 Oe-1) values ranging between 0.10 to 0.40. Overall, the change in the magnetization is minimal varying between values 0.20 and 0.25. This further commensurates our analysis that system 2 displays a quasi-antiferromagnetic ordering. Since the scale bar is less the magnetization moment values look scattered.
Both the ZFC/FC and M vs H measurements were conducted twice, and the data collected is averaged with respective the measurement number of iterations. Therefore, avoiding any discrepancies in the final data.
Q5. Investigations of additional experimental techniques like XPS or EPR should be performed for your samples.
- Our aim of the work is to study the magnetic bistability of individual SMM that can be retained to maximum when they are attached to a carbon nanotube surface. The same has been described in the manuscript. Furthermore, we have also discussed how the magnetic properties change concerning the replacement of central metal ions in the Ln-EDTA. Therefore, we believe XPS and EPR would not be needed. However, we further plan to study these magnetic CNT hybrid materials using various measurement techniques including XPS, EPR, AC magnetization and low-temperature electrical transport.
Q6. Conclusions should be rewritten in more informative way.
- The conclusion has been paraphrased and extended such that readers can understand it with ease.
New References:
- Moreno-Da Silva, S.; Martínez, J.I.; Develioglu, A.; Nieto-Ortega, B.; De Juan-Fernández, L.; Ruiz-Gonzalez, L.; Picón, A.; Oberli, S.; Alonso, P.J.; Moonshiram, D.; et al. Magnetic, Mechanically Interlocked Porphyrin-Carbon Nanotubes for Quantum Computation and Spintronics. J Am Chem Soc 2021, 143, 21286–21293, doi:10.1021/jacs.1c07058.
- Chen, J.S.; Trerayapiwat, K.J.; Sun, L.; Krzyaniak, M.D.; Wasielewski, M.R.; Rajh, T.; Sharifzadeh, S.; Ma, X. Long-Lived Electronic Spin Qubits in Single-Walled Carbon Nanotubes. Nat Commun 2023, 14, doi:10.1038/s41467-023-36031-z.
- Yin, X.; Deng, L.; Ruan, L.; Wu, Y.; Luo, F.; Qin, G.; Han, X.; Zhang, X. Recent Progress for Single-Molecule Magnets Based on Rare Earth Elements. Materials 2023, 16.
- De Oliveira Maciel, J.W.; Lemes, M.A.; Valdo, A.K.; Rabelo, R.; Martins, F.T.; Queiroz Maia, L.J.; De Santana, R.C.; Lloret, F.; Julve, M.; Cangussu, D. Europium(III), Terbium(III), and Gadolinium(III) Oxamato-Based Coordination Polymers: Visible Luminescence and Slow Magnetic Relaxation. Inorg Chem 2021, 60, 6176–6190, doi:10.1021/acs.inorgchem.0c03226.
- Ing, X.-L.; Hai, Y.-Q.; Han, T.; Chen, W.E.-P.; Ding, Y.-S.; Heng, Y.-Z. ALocal D 4h Symmetric Dysprosium(III) Single-Molecule Magnet with an Energy Barrier Exceeding 2000 K**. Chemistry A European Journal 2021, 27, 2623–2627, doi:10.26434/chemrxiv.12696578.v1.
- Han, T.; Giansiracusa, M..; Li, Z.-H.; Ding, Y.-S.; Chilton, N.; Winpenny, R.E.P.; Heng, Y.-Z. Exchange-Biasing in a Dinuclear Dysprosium(III) Single-Molecule Magnet with a Large Energy Barrier for Magnetisation Reversal**. Chemistry- A European Journal 2020, 26, 6773–6777, doi:10.26434/chemrxiv.10260494.v1.
- Katoh, K.; Sato, J.; Nakanishi, R.; Ara, F.; Komeda, T.; Kuwahara, Y.; Saito, T.; Breedlove, B.K.; Yamashita, M. Terbium(III) Bis-Phthalocyaninato Single-Molecule Magnet Encapsulated in a Single-Walled Carbon Nanotube. J Mater Chem C Mater 2021, 9, 10697–10704, doi:10.1039/d1tc01026c.
- Hymas, K.; Soncini, A. Origin of the Hysteresis of Magnetoconductance in a Supramolecular Spin-Valve Based on a TbPc2 Single-Molecule Magnet. Phys Rev B 2020, 102, doi:10.1103/PhysRevB.102.125310.

Round 2
Reviewer 3 Report
Authors make proper corrections according to reviewer remarks and I suggest to publish the paper as it is.